# Upper Respiratory Tract Disease in a Dog Infected by a Highly Pathogenic Avian A/H5N1 Virus

**DOI:** 10.3390/microorganisms12040689

**Published:** 2024-03-29

**Authors:** Olga Szaluś-Jordanow, Anna Golke, Tomasz Dzieciątkowski, Michał Czopowicz, Michał Kardas, Marcin Mickiewicz, Agata Moroz-Fik, Andrzej Łobaczewski, Iwona Markowska-Daniel, Tadeusz Frymus

**Affiliations:** 1Department of Small Animal Diseases with Clinic, Institute of Veterinary Medicine, Warsaw University of Life Sciences-SGGW, Nowoursynowska 159c, 02-776 Warsaw, Poland; tadeusz_frymus@sggw.edu.pl; 2Department of Preclinical Sciences, Institute of Veterinary Medicine, Warsaw University of Life Sciences (SGGW), Ciszewskiego 8, 02-786 Warsaw, Poland; anna_golke@sggw.edu.pl; 3Chair and Department of Medical Microbiology, Medical University of Warsaw, Chałubińskiego 5, 02-004 Warsaw, Poland; tdzieciatkowski@wum.edu.pl; 4Division of Veterinary Epidemiology and Economics, Institute of Veterinary Medicine, Warsaw University of Life Sciences-SGGW, Nowoursynowska 159c, 02-776 Warsaw, Poland; michal_czopowicz@sggw.edu.pl (M.C.); marcin_mickiewicz@sggw.edu.pl (M.M.); agata_moroz@sggw.edu.pl (A.M.-F.); iwona_markowska_daniel@sggw.edu.pl (I.M.-D.); 5Veterinary Clinic Auxilium, Arkadiusz Olkowski, Królewska Str. 64, 05-822 Milanówek, Poland; mkardas94@gmail.com (M.K.); alobaczewski007@gmail.com (A.Ł.)

**Keywords:** highly pathogenic avian influenza (HPAI), influenza virus type A (IAV), A/H5N1, dog

## Abstract

In summer 2023, during an outbreak of highly pathogenic avian influenza (HPAI) in cats in Poland, a 16-year-old dog was presented to the veterinary clinic with persistent, debilitating, dry cough, submandibular lymphadenomegaly, mild serous nasal discharge, and left apical heart murmur. A preliminary diagnosis of kennel cough was made and the treatment with amoxicillin/clavulanic acid and dexamethasone was initiated. Due to the lack of improvement within 2 days, a blood check-up, thoracic radiography and ultrasonography, and echocardiography were performed. Moreover, a rapid test for orthomyxovirus type A antigen in a throat swab was carried out and proved positive. The result was verified using RT-qPCR, which yielded a positive result for A/H5N1 influenza virus and negative results for A/H1N1, A/H3N2, type B influenza, and SARS-CoV-2. This case indicates that HPAI should be considered as a differential diagnosis not only in cats, but also in dogs with upper respiratory tract disease, particularly in regions experiencing A/H5N1 avian influenza outbreaks.

## 1. Introduction

Influenza, a highly contagious upper respiratory tract infection caused by Orthomyxoviruses, has emerged as a health concern, both in canine companions and other canids. Several subtypes of influenza virus type A (IAV) have been identified in canids, and the strains were originally associated with different host species. These included A/H1N1 virus, predominantly affecting humans and swine, A/H3N2, A/H3N6, A/H5N2, A/H9N2, and A/H10N8, originally linked to birds, and A/H3N8, originating from equine influenza virus [1,2,3,4,5,6,7]. Only A/H3N8 and A/H3N2 subtypes became adapted to dogs as canine influenza virus (CIV). The first one transferred in the USA from horses to dogs was circulating since the early 2000s at first among racing greyhounds in Florida, and then spread to other breeds and regions in the USA [2,8] before it probably went extinct around 2016 [9]. Similar transmission of this equine virus to dogs occurred during an equine influenza epidemic in Australia in 2007 [10]. By 2006 the A/H3N2 virus, most probably of avian origin, was adapted to dogs in China or Korea [11], where it induced several endemic canine outbreaks, and was later transmitted to the USA [12].

Both A/H3N8 and A/H3N2 CIVs can cause an acute upper respiratory tract infection with fever, coughing, and nasal discharge; however, only a few affected dogs develop severe disease. Though secondary bacterial infections may result in pneumonia or other complications, most patients recover after 1 to 3 weeks, and the fatality rate is usually <1% [13]. Therefore, these agents, together with “seasonal” IAVs of humans and a variety of other mammalian and avian species, are classified as “low pathogenic” viruses. In contrast, emerging almost exclusively in birds, “highly pathogenic” IAVs can induce a severe, generalized infection affecting not only the lungs but also many other organs, resulting in mass mortality of both poultry and wild birds. The most important highly pathogenic subtype is the A/H5N1 virus that emerged in 1996 in Southeast Asia and at the beginning of the 21st century, which induced a large and long-lasting epidemic in that region [14]. Subsequently, this epidemic extended to Africa, Europe, and recently to the Americas [15]. On very rare occasions mammals were also affected, including over 870 humans, with a fatality rate of more than 50% [16]. Feeding with fresh or frozen contaminated carcasses, hunting, or other contacts with birds resulted in sporadic, but usually very severe cases both in companion animals [2,17,18,19,20,21], as well as in wild cats [19]. However, although the A/H5N1 virus has been persistently circulating in birds of Asia, Europe, and Africa for over 20 years, feline cases of highly pathogenic avian influenza (HPAI) have been very rare so far, and usually isolated. Mass outbreaks in only two feline Korean shelters were noted in 2023, and the most probable reason for this was contaminated duck meat contained in cat food [22]. Similarly, in several regions of Poland, in the summer of 2023, numerous cases of HPAI in owned domestic cats occurred, involving both outdoor and indoor animals [23,24]. Again, contaminated poultry meat could not be ruled out [23,25]. At least 34 feline cases including a captive caracal were officially confirmed (Chief Vet. Off., https://www.wetgiw.gov.pl/main/komunikaty/Komunikat-VIIGLW-w-sprawie-chorobykotow/idn:2302, accessed on 11 January 2024). However, the true number of affected cats was higher; at least in our laboratory several additional feline cases were diagnosed, along with four infections in ferrets and one in a sick dog [26]. There has been evidence indicating that dogs can also harbor the A/H5N1virus asymptomatically [2,27]. However, in contrast to cats, spontaneous disease connected with an A/H5N1 infection has hardly been described in dogs, thus far. Therefore, the goal of this case report is to present the clinical course, lung ultrasound examination, and other findings in a canine patient affected by the A/H5N1 virus during the cluster of HPAI feline cases in summer 2023 in Poland.

## 2. Materials and Methods

### 2.1. Patient Description

A crossbred, intact male dog, around 16 years of age, weighing 7 kg, was presented to the veterinary clinic on June 2023 due to persistent, dry cough, accompanied by mild serous nasal discharge, with no other symptoms. The dog had been adopted from a rural area just a week prior to the visit, with no available information regarding its previous health, living conditions, and dietary history. After adoption, the dog lived in a small town with an approximate population of 16,000 residents. During the week after adoption the dog was fed dry food, no fresh or frozen meat was used. In the district where the dog resided, no infections were reported either in wild or domestic birds. The nearest confirmed infections of A/H5N1 in wild birds were less than a month prior to the onset of symptoms, at locations approximately 200 and 270 km from the dog’s residence https://www.wetgiw.gov.pl/nadzor-weterynaryjny/hpai (accessed on 27 August 2023). 

### 2.2. Blood Analysis

Venous blood was collected into a serum clot tube and a tube with EDTA, and sent to a veterinary laboratory for hematological and biochemical analyses.

### 2.3. Ultrasound and X-ray Examination

Ultrasound examination of the lungs and heart was performed using a Mindray M9 (Shenzhen 518057, China) with a 5-1s MHz phased array transducer for heart examination and 12-4s MHz linear transducer for lung examination. X-ray examination of the chest was performed in two projections using an Ecoray-1060HF Portable X-ray Generator (Ecoray, Seoul, Republic of Korea).

### 2.4. Viral and Serogical Tests

Throat swabs for virological testing were collected during clinical examination. Initial screening of the swabs was performed using the Fluorecare^®^ (Milapharm, Ruislip, UK) test for detection of the orthomyxovirus type A antigen in humans. Next, molecular diagnostics for IAV subtypes H1N1, H3N2, H5N1, influenza virus type B (IBV), and SARS-CoV-2 were performed. RNA was extracted from an additional throat swab using a Total RNA Mini Kit (A&A Biotechnology, Gdańsk, Poland), according to the manufacturer’s instructions. One-step reverse transcription real-time PCR (RT-qPCR) was performed using the CFx96 system (BioRad, Hercules, CA, USA). Real-Time Multiplex RT-PCR Kit (LifeRiver, San Diego, CA, USA) for SARS-CoV-2 detection and with an in-house method described by Stefańska et al. for IAVs [28] RT-qPCR results with a quantification cycle (Cq) of ≤35.00 were considered positive.

## 3. Results

### 3.1. Clinical Course of the Disease and Treatment

During the clinical examination, severe, dry cough was present, and according to the owner, there were many coughing episodes throughout the day. The submandibular lymph nodes were enlarged. The body temperature was 37.7 °C, 130 heartbeats per minute, 24 breaths per minute, mucous membranes were moist and pink, capillary refill time < 2 s, during auscultation left apical heart murmur, grade II/VI was detected, but no pathological lung sounds were found. Based on the clinical examination, kennel cough was the preliminary diagnosis, and treatment with 10 mg amoxicillin and 2.5 mg clavulanic acid per kg body weight (b.w.) (Clavaseptin 250 mg tabl., Vetoquinol Biowet, Gorzów Wielkopolski, Poland) twice daily orally was started as well as dexamethasone (dexamethasone sodium phosphate 1.32 mg/mL, dexamethasone phenylpropionate 2.67 mg/mL—Dexafort, Intervet International B.V., Boxmeer, The Netherlands) 0.05 mL/kg body weight (b.w.) subcutaneously (s.c.). However, after two days of treatment there was no improvement in the patient’s clinical state. The debilitating, dry cough persisted. In response to this clinical challenge, further diagnostic investigations were undertaken, including a broad blood examination, chest radiography (X-ray), chest ultrasonography, and echocardiographic examination. In anticipation of the results of the RT-PCR for influenza, after 4 days of ineffective treatment the amoxicillin with clavulanic acid was replaced by marbofloxacin 2 mg/kg b.w. s.c. once daily (Marbovet, VET-AGRO, Lublin, Poland) with cefovecin (Convenia, Zoetis, Warsaw, Poland) 8 mg/kg bw s.c. As an antitussive, butorphanol was prescribed in a dose of 0.2 mg/kg b.w. i.m. three times per day (Torphadine, Dechra, Cheshire, UK). The intensity of coughing decreased, but the serous nasal discharge became purulent. The entire course of the therapy spanned 18 days, after which the cough ultimately subsided. The patient returned to full health following the A/H5N1 infection.

### 3.2. Blood Analysis

In the blood chemistry, the activity of alkaline phosphatase (AP) was 382 U/L, with an reference interval (RI) of 0 to 160 U/L, the total protein concentration was 7.99 g/dL (RI 5.4 to 7.5 g/dL), and the globulin level was 49.3 g/L (RI 25 to 45 g/L). The values of urea and creatinine were within the RI. The blood cell count was unremarkable, except for the hematocrit value 36.6% (RI 37 to 55%) and erythrocyte number 5.16 M/µL (RI 5.5 to 8.5 M/µL).

### 3.3. X-ray and Ultrasound Examination

No pathological changes were detected in the lung parenchyma image in the X-ray (Figure 1 and Figure 2). The ultrasound examination revealed in all views a normal, aerated lung pattern. The pleural line was intact, and A-line artifacts were present (Figure 3). The sliding sign was also observed. Echocardiography revealed a mild thickening of the mitral valve leaflets and a slight regurgitation (Figure 4) of this valve. The size of the heart chambers was within the normal range, and myocardial contractility was normal.

## 4. Discussion

Recent epizootics of HPAI in birds have also led to an increase in reported carnivore A/H5N1 infections over the last few years in Europe. Some examples are cases in cats in Poland and France [23,24,29], in foxes [30,31], in American minks, raccoon dogs on several fur farms in Finland [32], and in minks in Spain [33]. Generally, this infection has been much better characterized in domestic cats and wild felines than in other carnivores. However, in the literature, there is a growing body of evidence indicating that both domestic dogs and wild canids can also harbor this agent, with the potential for both asymptomatic shedding and clinical disease. The few experimental infections of dogs with A/H5N1 virus performed so far resulted only in subclinical replication of the agent or a transient fever and respiratory symptoms, which were rather mild [34,35]. However, it should be emphasized that the outcomes of natural infections can significantly differ from experimental infections, as evidenced by studies on foxes. A natural A/H5N1 infection has been recently confirmed in two deceased foxes in Italy [31]. The animals were found in an area experiencing mass HPAI mortalities among farmed pheasants, pointing toward a possible link between avian canid infections. Post-mortem examinations revealed in these foxes lung alterations and the presence of bloody fluid in the pleural cavity. Further insight into the virus’s impact on foxes was provided by an experimental study conducted by Reperant et al. [36]. The animals were divided into two groups: one received the A/H5N1 virus intratracheally, while the other was exposed through the consumption of infected bird carcasses. Foxes from the first group developed severe respiratory and systemic disease, including pneumonia, myocarditis, and encephalitis. In contrast, the group that consumed infected carcasses excreted the virus without manifesting severe illness, suggesting a potential asymptomatic shedding by foxes. The geographical scope of wild canid HPAI infections extends beyond Europe. In Asia, the A/H5N1 virus was confirmed in an Ezo red fox (*Vulpes vulpes schrencki*) and a tanuki (*Nyctereutes procyonoides albus*). The Ezo red fox was presented with viral meningoencephalitis and moderate virus replication in the upper respiratory tract [21]. 

In contrast to wild canids, reports about natural HPAI virus infections in domestic dogs have been scarce, thus far. There is only one published report of a natural A/H5N1 infection in a one-year-old dog from Thailand. This dog, which presented with severe respiratory symptoms, died in 2004 after ingesting a duck infected with the A/H5N1 virus. The virus was detected in multiple organs [2]. Controlled infection studies confirmed that dogs are susceptible to the A/H5N1 virus and can develop respiratory symptoms [34,35,37]. Our patient was a sick domestic dog, positive both in the influenza type A antigen test and in PCR for A/H5N1 sequences, presented to the clinic during a cluster of highly deadly cases in domestic cats caused by the A/H5N1 virus [23,24,25]. The dog showed symptoms that were initially suggestive of severe kennel cough, characterized by enlarged submandibular lymph nodes and a dry cough that persisted despite conventional treatment for an 18-day course. Unfortunately, information about the diet of this dog before the onset of the symptoms was not available. 

The case described in our report confirms that on rare occasions the A/H5N1 virus can also induce a natural severe respiratory disease in dogs. While in some of them the infection remains asymptomatic, capable of shedding the virus [35], others exhibit mild symptoms such as transient fever [34], or even fatal disease [20]. Recently, five seropositive healthy dogs were found on a farm in Italy during an outbreak of clade 2.3.4.4b HPAI A/H5N1 infection in poultry. The virus isolated from birds had in the PB2 gene the T271A mutation, which is a marker of virus adaptation to mammals [38]. Studies by Maas et al. [35] using labeled A/H5N1 virus highlighted the agents’ ability to adhere to tissues in both the upper and lower respiratory tracts of dogs. This suggests a potential role of dogs as intermediate hosts in transmitting HPAI from birds to humans, as discussed by Chen et al. [37]. 

In Poland, as in most European countries, dogs presenting with respiratory symptoms are not routinely tested for influenza. The presented case confirms that in Europe dogs can develop a natural disease induced by the A/H5N1 virus, since HPAI outbreaks have become common in Europe during the last several years. Thus, in areas with endemic HPAI in birds, this infection should be considered in the differential diagnostics not only in cats but also in dogs, especially in patients with respiratory disease. 

## Figures and Tables

**Figure 1 microorganisms-12-00689-f001:**
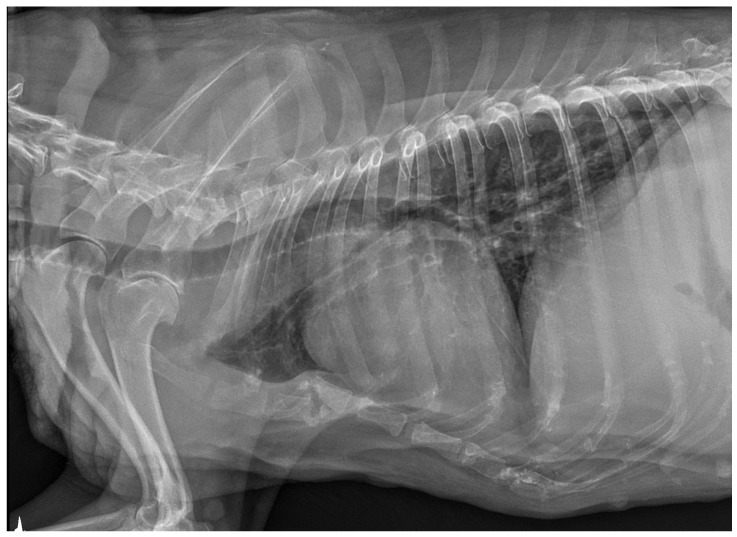
The thoracic X-ray image in the lateral right–left (RL) position shows no pathological alterations within the lung parenchyma and cardiac silhouette. Slight elevation of the trachea in the precordial area is visible.

**Figure 2 microorganisms-12-00689-f002:**
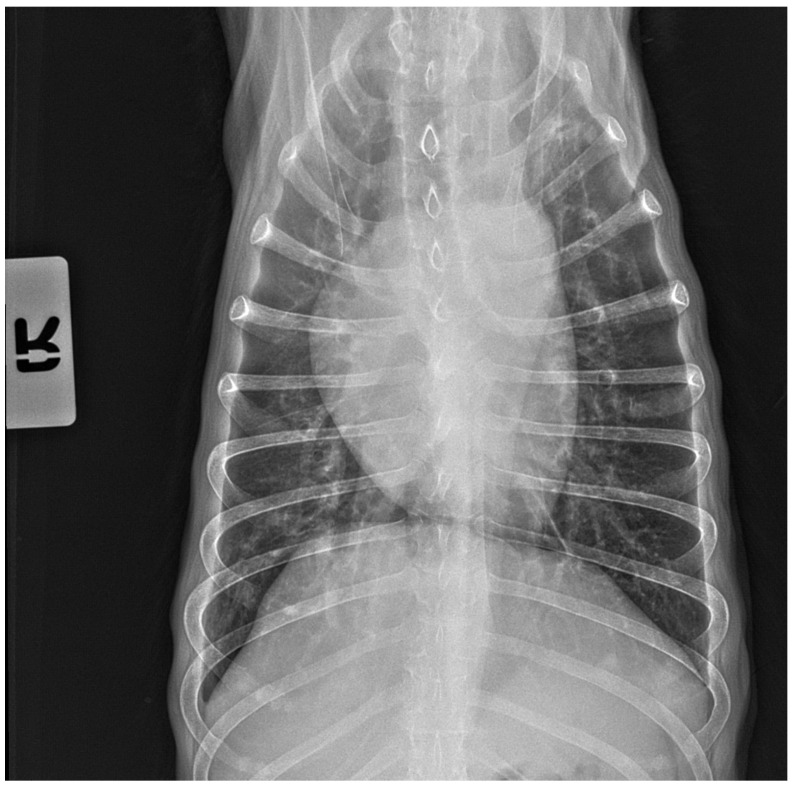
The thoracic X-ray image in the anteroposterior (AP) position shows no pathological alterations within the lung parenchyma and cardiac silhouette.

**Figure 3 microorganisms-12-00689-f003:**
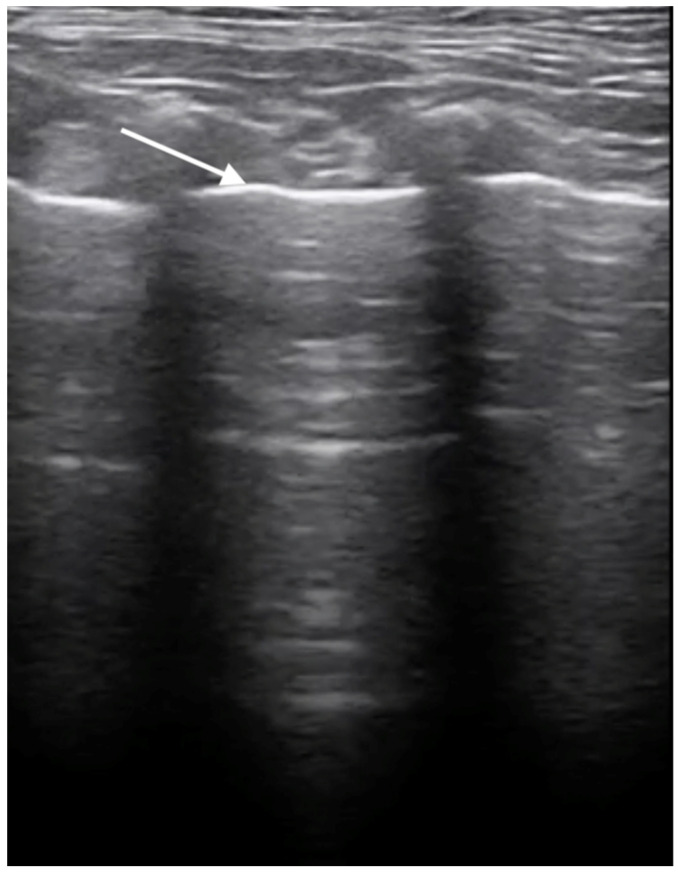
Ultrasonographic examination of the lungs. Image of a healthy lung filled with air; the white arrow indicates the normal, smooth pleural line.

**Figure 4 microorganisms-12-00689-f004:**
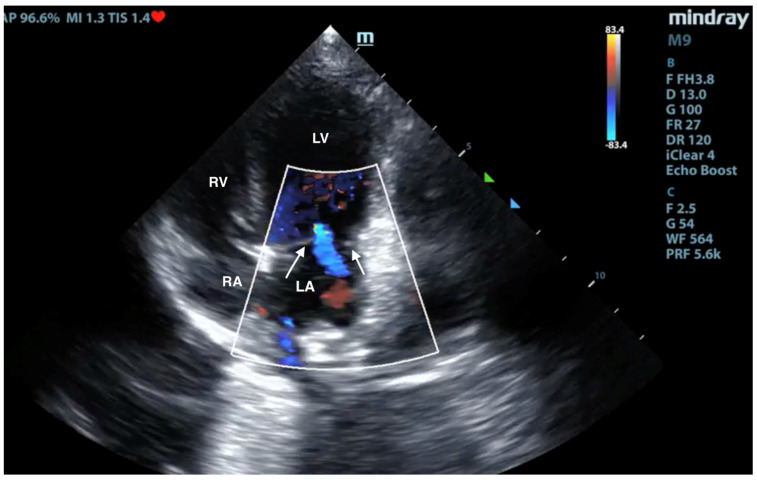
Scan from phased array probe obtained during echocardiography examination performed by a board-certified echocardiography specialist. Mild thickening of the mitral valve and slight regurgitation are visible. The white arrows indicate mitral valve leaflets. LA—left atrium, LV—left ventricle, RA—right atrium, RV—right ventricle.

## Data Availability

No new data were created or analyzed in this study. Data sharing is not applicable to this article.

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
