# Peer review of "Upper Respiratory Tract Disease in a Dog Infected by a Highly Pathogenic Avian A/H5N1 Virus"

_microorganisms, 2024, doi:10.3390/microorganisms12040689_

Round 1
Reviewer 1 Report
Comments and Suggestions for Authors
This manuscript is highly significant in that it diagnosed H5N1 HPAI virus infection in a domestic dog, which could have been overlooked.
1. Overall, there is insufficient information about this patent dog.
- In which area is the veterinary clinic that the patient's dog visited? Is it a big city?
- Are the rural areas where dogs lived before adoption reported HPAI outbreaks in wild birds or poultry?
- What led you to perform an AIV infection test on a dog that showed no abnormal symptoms on X-ray or ultrasound?
- What ultimately happened to this patient dog? Did it recover? In how many days? Did it die?
- Are there other mammals and birds that are epidemiologically related? Have you done a test of HPAI infection?
2. To write the Discussion part, it is recommended to think more about the context.
- The first and second paragraphs discuss 'experiment' vs 'natural' infection, 'wild' vs 'domestic' animals. However, it is not clear what and why the authors are trying to claim.
- Lines 181-187, the results of the two-group experiment by Referant et al. are cited in detail. How does citing that experimental results help your paper's argument? I'm afraid your discussion is missing such point.
Author Response
This manuscript is highly significant in that it diagnosed H5N1 HPAI virus infection in a domestic dog, which could have been overlooked.
Reviewer: 1. Overall, there is insufficient information about this patent dog.
- In which area is the veterinary clinic that the patient's dog visited? Is it a big city?
Our answer: added
Reviewer: - Are the rural areas where dogs lived before adoption reported HPAI outbreaks in wild birds or poultry?
Our answer: added
Reviewer: What led you to perform an AIV infection test on a dog that showed no abnormal symptoms on X-ray or ultrasound?
Our answer: Thank you for your inquiry regarding the testing for AIV infection in our canine patient. We appreciate your review and your insightful questions. In addition to the outbreak of H5N1 influenza among cats in Poland, which we mentioned in our manuscript, there were also cases in ferrets—unpublished data—that contributed to our decision to conduct an AIV test. Furthermore, the patient exhibited severe and persistent coughing that was unresponsive to treatment, which was an atypical presentation and prompted us to proceed with the AIV testing. This line of reasoning is delineated within the section outlining the clinical course and lung ultrasound examination findings
Reviewer: What ultimately happened to this patient dog? Did it recover? In how many days? Did it die?
Our answer: The patient's condition returned to normal after 18 days, having survived the A/H5N1 infection. For greater clarity, we have added an additional sentence in line 143-144.
Reviewer: Are there other mammals and birds that are epidemiologically related? Have you done a test of HPAI infection?
Our answer: We appreciate your question regarding epidemiological connections to other mammals and birds. Currently, we do not have knowledge of direct contact between our patient and other animals or birds infected with A/H5N1. As stated earlier in the added manuscript section, the nearest confirmed cases of avian infection were 200-270 km away from the dog's residence. Additionally, none of the authors presented clinical symptoms and therefore were not tested for influenza.
- To write the Discussion part, it is recommended to think more about the context.
Reviewer: The first and second paragraphs discuss 'experiment' vs 'natural' infection, 'wild' vs 'domestic' animals. However, it is not clear what and why the authors are trying to claim.Lines 181-187, the results of the two-group experiment by Referant et al. are cited in detail. How does citing that experimental results help your paper's argument? I'm afraid your discussion is missing such point.
Our answer: Thank you for your comments. In light of the fact that infections in cats and felids, which represent a much better understood phenomenon than those in dogs, have been widely documented, we felt it was crucial to emphasize that dogs and canids are also susceptible. Given that natural, fatal infections in dogs have only been described in a single publication worldwide, we decided to include information on experimental infections and their significance in canines. Notably, as observed in foxes, the course of natural infection differs from experimental infection where the virus is administered tracheally. We have added additional information in the manuscript that, in our opinion, better highlights these points. We believe this will provide a clearer understanding of the distinct nature of natural versus experimental infections in canids and the relevance of these findings to domestic dogs.

Reviewer 2 Report
Comments and Suggestions for Authors
The manuscript titled "Upper respiratory tract disease in a dog infected by a highly pathogenic avian A / H5N1 virus" is a case report on a domestic dog in Poland that was symptomatic for a URI and tested positive for A/H5N1, an emerging HPAIV. This is an interesting case report and clearly emphasizes the need to add influenza as a differential for mammals presenting with URI and are in areas where H5N1 is circulating especially in wild bird populations. I have only minor comments to share:
Line 83: Please correct "crossbread" to crossbred.
Line 83: Was the dog intact or neutered?
Line 87: If information available, please add details on what food was fed to the dog.
Line 124: b.w. s.c. should be defined.
Author Response
Reviewer: The manuscript titled "Upper respiratory tract disease in a dog infected by a highly pathogenic avian A / H5N1 virus" is a case report on a domestic dog in Poland that was symptomatic for a URI and tested positive for A/H5N1, an emerging HPAIV. This is an interesting case report and clearly emphasizes the need to add influenza as a differential for mammals presenting with URI and are in areas where H5N1 is circulating especially in wild bird populations. I have only minor comments to share:
Our answer: Thank you for your insightful comments and the time you have invested in reviewing our manuscript. We are pleased to hear that you found the case report interesting and appreciate your highlighting the importance of considering influenza in the differential diagnosis in dog.
Reviewer: Line 83: Please correct "crossbread" to crossbred.
Our answer: done
Reviewer: Line 83: Was the dog intact or neutered?
Our answer: added
Reviewer: Line 87: If information available, please add details on what food was fed to the dog.
Our answer: added
Reviewer: Line 124: b.w. s.c. should be defined.
Our answer: done
